# Land Suitability Evaluation and an Interval Stochastic Fuzzy Programming-Based Optimization Model for Land-Use Planning and Environmental Policy Analysis

**DOI:** 10.3390/ijerph16214124

**Published:** 2019-10-25

**Authors:** Zuo Zhang, Min Zhou, Guoliang Ou, Shukui Tan, Yan Song, Lu Zhang, Xin Nie

**Affiliations:** 1Collage of Public Administration, Central China Normal University, Wuhan 430079, China; zhangzuocug@163.com; 2College of Public Administration, Huazhong University of Science and Technology, Wuhan 430074, China; tansk@126.com (S.T.); zhanglu54522@126.com (L.Z.); 3School of Construction and Environmental Engineering, Shenzhen Polytechnic, Shenzhen 518055, China; 4The Department of City and Regional Planning, The University of North Carolina at Chapel Hill, Chapel Hill, NC 27599, USA; ys@email.unc.edu; 5School of Public Administration, Guangxi University, Nanning 530004, China; TOEFL678@163.com

**Keywords:** land-use allocation, interval stochastic fuzzy programming, land suitability evaluation, hybrid quantitative and spatial optimization model, environmental constraints

## Abstract

People explosion and fast economic growth are bringing a more serious land resource shortage crisis. Rational land-use allocation can effectively reduce this burden. Existing land-use allocation models may deal with a lot of challenges of land-use planning. This study proposed a hybrid quantitative and spatial optimization land-use allocation model that could enrich the land-use allocation method system. This model has three advantages compared to former methods: (1) this model can simultaneously solve the quantitative land area optimization problem and spatial allocation problem, which are the two core aspects of land-use allocation; (2) the land suitability assessment method considers various geographical, economic and environmental factors which are essential to land-use allocation; (3) this model used an interval stochastic fuzzy programming land-use allocation model to solve the quantitative land area optimization problem. This model not only considers three uncertainties in the natural system but also involves various economic, social, ecological and environmental constraints—most of which are specifically put into the optimization process. The proposed model has been applied to a real case study in Liannan county, Guangdong province, China. The results could help land managers and decision makers to conduct sound land-use planning/policy and could help scientists understand the inner contradiction among economic development, environmental protection, and land use.

## 1. Introduction

Due to population expansion and rapid economic growth, land resource is becoming more valuable in China and most developing countries. However, the land area of every country is limited; it is impossible to enlarge the land area, except that people migrate to other areas to live. The only way to solve this problem is to conduct sound land-use planning. Land-use allocation is the key point of land-use planning. Scientific land-use allocation could promote sustainable development all over the world.

Rational land-use allocation involves two essential problems: quantitative land area optimization and spatial optimization. In previous studies, different disciplines and methods have been used to try to solve the land-use allocation problem. In these exploratory endeavors, mathematical models play a vital role in quantitative analysis and decision making. Based on the review and sorting of existent related literature, these models could be divided into the following categories: first, spatial optimization models [1,2,3,4,5,6,7,8,9,10,11,12,13,14]—the core problem of which lies in how to properly adjust and control the spatial relations among land-use resources so as to make their spatial structure reaches an optimal status which best meets the requirement of a sustainable development target. Second, programming models [15,16,17,18,19,20,21,22], including linear programming models, multi-objective programming models, and non-linear programming models, generally apply mathematical models to achieve the best outcome such as maximum profit or lowest cost. Third, simulation models [12,23,24,25,26,27,28,29,30,31,32], which provide the process of creating and analyzing a digital land-use prototype of a model to predict its performance in the real world. Forth, intelligent models [8,33,34,35,36,37,38,39,40,41], which is a class of computational models, such as the agent-based model (ABM) and the cellular automata model. Models in this category can evaluate their impact by simulating individual land-use agents’ actions and their interactions, enabling knowledge of the overall system characteristics.

In most of the above methods, a geographic information system (GIS) has been employed in different degrees, which provide a solid foundation for spatial data of land-use storing and analyzing. Further, a number of mathematical models built upon the foundation of geospatial interactions different land-use with various spatial scales are included [42]. In addition, the scope of the spatial scale generally determines the framework of model [21,22]. Each of the above models has its strengths, such as explicating the land-use structure, handling the optimizing land-use management or predicting land-use future trends. However, exciting models also have some deficiencies such as the assumptions of the models are strict and have poor adaptation to the variation of practical conditions. Moreover, the accuracy of the model results is always affected by uncertain or big-scale factors of real-world environment. Compared with these models, uncertain mathematical models go forward with small limitations of assumptions and conditions and could determine how likely certain outcomes are if some factors are not exactly known. Uncertain models have been effectively applied in simulating and optimizing resource management problems, such as water resource management [43,44], plant resource management [45], energy management [46,47], and so on. Nevertheless, insufficient consideration is given to uncertainties and environment-affecting factors in land-use allocation models [20,22], and an uncertain model is rarely used in current studies of land-use allocation problems.

In this study, we combine a spatial optimization method and an uncertain model into the same framework to solve a land-use allocation problem. This model is based on a spatial land suitability assessment method and an interval stochastic fuzzy programming model. It has three main advantages compared to previous methods: first, the model can simultaneously solve the quantitative land area optimization problem and spatial allocation problem which are the two core aspects of land-use allocation; second, the land suitability assessment method considers various geographical, economic and environmental factors which are essential to land-use allocation; third, the model used an interval stochastic fuzzy programming land-use allocation model to solve the quantitative land area optimization problem. This model not only considers three uncertainties in the natural system but also involves various economic, social, ecological and environmental constraints—most of which are specifically put into the optimization process. The proposed model has been applied to a real case study in Liannan county, Guangdong province, China. The results could help land managers and decision makers to conduct sound land-use planning/policy and could help scientists understand the inner contradiction among economic development, environmental protection, and land use.

## 2. The Study Area

Liannan, a county in Qingyuan city, is located in 112°2′2′′~108°38′ E, 24°17′16′′~24°56′2′′ N, China (Figure 1). The area of Liannan is 1240.91 km^2^, occupying 0.69% of the land area of Guangdong province, China. Liannan contains seven towns (Damaishan, Daping, Huoshui, Sanjiang, Sanpai, Xiangping, and Zhaigang) and the population is approximately 170,791 (2015). Due to the fact that Liannan is an ethnic minority autonomous county, its residents are mostly Yao minority. Liannan’s gross domestic product (GDP) increased rapidly from 1.82 × 10^9^ CNY in 2000 to 3.59 × 10^9^ CNY in 2015, in concert with a population increase from 0.164 million to 0.171million. Liannan is a subtropical monsoon climate, with an annual average temperature of 19.5 °C, and an average annual rainfall of 1660.5 mm. Liannan’s terrain is long and narrow in north to south, where there are mostly mountainous areas and little rivers and streams.

According to the results of the second national land survey of Liannan, the total land area of the county is 124,091.46 hm^2^, the total land area is 118,846.55 hm^2^, and the land-use rate is 95.77%. Liannan’s agricultural land is 114,975.02 hm^2^ (89.55% are forest land), accounting for 92.65% of the total land area; and construction land is 3871.53 hm^2^, accounting for 3.12% of the total land area (Figure 2). In general, land-use in Liannan has the following characteristics and problems: first, natural conditions of land-use are poor. Liannan is located in the mountainous region, with steep terrain and poor soil. Second, Liannan’s average per person arable land area is scarce (0.063 hm^2^), which is lower than the average level of Qingyuan city (0.071 hm^2^). Third, the total land-use rate of Liannan is high, reaching 95.77%, and accounting for 92.65% of the total land area that has been used as agricultural land—of which, the vast majority is forest land. Forth, while the population of Liannan is growing, and the cultivated land continues to decrease, reserve cultivated land resources are poor in both quantity and quality.

At present, Guangdong province implements the new strategies of the Pan-Pearl River Delta regional cooperation [48] and the coordinated development of mountainous areas. Further, Liannan implements the “eco-economic” development strategy for the Yao minority autonomous county. All of these designate Liannan’s land-use direction for the future. According to Liannan land-use planning (2010–2020), authorized by the Guangdong province in 2010, the following goals were put forward: first, based on the macro-control functions of land-use planning, coordinate the land resources demand from different industries, restructure the industrial layout and establish of the overall spatial layout model that adapting to regional socio-economic development. Second, improve the rationality of land-use structure adjustment, protect primary farmland and guarantee the reasonable needs of construction lands for economic and social development. Third, strengthen ecological protection, improve the ecological environment and land-use efficiency, and establish “eco-economic” urban and rural construction land-use model.

## 3. Land Suitability Evaluation

### 3.1. GIS-Based Land Suitability Evaluation Model

Land suitability evaluation as one of the foundations of the land-use management process [49], can not only determine the spatial pattern of future land-use in the study area but can also reduce environmental quality degradation under the premise of guaranteeing the maximum economic benefit [18]. The land suitability evaluation could make a quantitative assessment for a particular land-use in the study area, which relates to multi-attributes including physics, location or institution [50]. The evaluating results of suitability level for each spatial unit in different land-use types could provide a scientific basis for the subsequent optimization of land-use allocation [51].

### 3.2. Suitability Evaluation Factors

The selection of a set of key factors including geography, economy, and environment is essential and important to land suitability evaluation [17]. The selection of evaluation factors is related to the collection and processing of large amounts of data—not only the most basic part of the land suitability assessment, but also the most important and time-consuming part of the overall evaluation process. Selection of evaluation factors requires comprehensive, representative, quantifiable measurements and source data available [52]. Referring to the indicators selected in previous studies [17,18,20], and considering the characteristics of the study area and data availability, key assessment factors for major land-use categories were selected and are listed as following (Table 1).

In the cultivated land category, suitability evaluating factors include four groups and eleven attributes. Topography, including elevation, slope gradient, and aspect, play an important role in the redistribution of hydrothermal conditions and the migration of matter. This is especially important for mountainous counties such as Liannan, which largely determines the difficulty and economic benefits of land development. Ground conditions, including soil organic matter, contains, soil thickness, topsoil texture, and soil PH value, determine the quality of the soil which is the most basic material of agricultural production. Geological hazards, the susceptibility of hazards and the distance to fault may influence or pose threats to agricultural production activities, which are especially important to mountainous areas. Hydrology conditions, including distance to water area and irrigation conditions, are crucial guarantees for crop growth.

In the commercial land, residential land, and industrial land cultivated category, suitability evaluating factors include four groups and nine attributes. Among them, factors of topography geological hazards are selected in keeping with the cultivated land category. Beyond that, in ground conditions and hydrology conditions, the factors affect the quality of cultivated soils, which may be related to the conversion of agricultural land to construction land. Further, spatial locations, including the distance to main roads and the distance to core towns, reflect the levels of accessibility and basic infrastructure.

The spatial distribution map for each factor is developed separately by employing a GIS tool, which can be convenient to calculate the terrain’s slope gradient and aspect, calculate and re-classify the various Euclidean distances of different spatial units.

### 3.3. Factor Weighting and Overlap Analysis

Factor weighting not only relates to the relative importance of each factor but also determines whether the results of the land suitability evaluation are scientific and reasonable. Here, the Delphi approach [53] is applied to weight each factor depending on its relative importance in the evaluation system based on fifteen experts’ knowledge. Further, the criterion values in each factor are standardized in order to adapt to five levels (S1, S2, S3, N1, N2) which assigned integer scores ranging from 100 to 0.

Based on overlap analysis in GIS, a set of layers with different factor values and criteria can be respectively formed to a single suitability map according to land-use categories (Figure 3 and Figure 4). In this process, a common weighted linear method [17,18] is used to calculate the results of land suitability through the following formula:(1)$$S_{j}=\sum_{i=1}^{n}W_{ij}x_{ij}$$
where *S_j_* represents the level of suitability to the land-use category *j*, *n* is the number of criteria related to category *j*, *W**_ij_* (0–1) represents the weight of criterion *i* of category *j*, and *X_ij_* represents scores of criterion *i* in category *j*. The values of *W_ij_* and *X_ij_* are presented in Table 1. Figure 5 respectively shows the final land suitability results for cultivated land (Figure 5a), commercial land, residential land, and industrial land (Figure 5b).

## 4. Interval Stochastic Fuzzy Programming and Land Suitability Evaluation Based Land-Use Allocation Model

A typical land-use allocation system may consider three types of factors: (1) Economic factors, such as GDP, investment, benefit, product, resource consumption, and so on; (2) Social factors, such as population, labor, and so on; (3) Ecological and environmental factors, such as wastewater, solid waste, waste gas, forest and grass coverage, soil erosion, fertilizer application, and so on. After analyzing these influence factors, we can conclude four characters of the land-use allocation system: (1) Systematicness. A land-use allocation system is an organic integrity; all factors of the integrity will interplay with each other; (2) Dynamic. All factors will change all the time with the natural and artificial variations; (3) Complexity. The influencing mechanism among these factors is very complicated; every factor will influence other factors, and feedback mechanisms could have many forms; (4) Uncertainty. Every factor is not certain, because the time, spatial location and external environment are always different.

Based on the above analysis and existing theoretical land-use allocation models, this study integrates an interval stochastic fuzzy programming model (present in Section 3) and a land suitability assessment model (present in Section 4) into one framework.

### 4.1. Objective Function

The objective function is the benefit from the land-use system, and it equals all benefits from cultivated land, garden land, forest land, grassland, construction land, transportation land and water land minus costs from all types of land use. In detail, benefit is from the industries on the lands. For example, benefit from cultivated land comes from all farmland industries (e.g., planting industry); benefit from construction land comes from all construction industries such as the coal industry, paper industry, textile industry, and so on. Cost is from the maintenance fees of all types of land use. For example, we must afford cultivated land and construction land with energy and water; furthermore, these two types of land use will generate wastewater, solid waste and air pollution, which are needed to handle. Based on the above settings, our objective function can be expressed as:(2)$$\begin{array}{ll} Max\;\mathrm{NB}L^{\pm } & \underset{\approx }{=}\sum_{k\;=\;1}^{4}\sum_{j=1}^{7}\left(b_{\;j\;,\;k}^{\pm }\;\times \;x_{\;j,\;k}^{\pm }\right)\\ & -\sum_{k\;=\;1}^{4}\sum_{j=1}^{7}\left[\left(\mathrm{UWT}C_{j,\;k}^{\pm }+\mathrm{UST}C_{j,\;k}^{\pm }+\mathrm{UGT}C_{j,\;k}^{\pm }+\mathrm{UWS}C_{j,\;k}^{\pm }+\mathrm{UES}C_{j,\;k}^{\pm }\right)\times x_{j,\;k}^{\pm }\right]\\ & -\mathrm{UM}C_{i,\;j=8\;}^{\pm }\times x_{i,\;j=8}^{\pm }-\mathrm{UD}C_{i,\;j\;=\;9}^{\pm }\;\times \;x_{i,\;j\;=\;9}^{\pm } \end{array}$$
where the detailed descriptions of the symbols are shown in Table 2.

### 4.2. Economic Constraints

(i) Government investment constraints

In Liannan county, all costs will be afforded by government investment, and so the government investment constraints can be expressed as:(3)$$\begin{array}{l} \sum_{k\;=\;1}^{4}\sum_{j=1}^{7}\left[\left(\mathrm{UWT}C_{j,\;k}^{\pm }+\mathrm{UST}C_{j,\;k}^{\pm }+\mathrm{UGT}C_{j,\;k}^{\pm }+\mathrm{UWS}C_{j,\;k}^{\pm }+\mathrm{UES}C_{j,\;k}^{\pm }\right)\times x_{j,\;k}^{\pm }\right]\\ +\mathrm{UM}C_{j=8\;}^{\pm }\times x_{j=8}^{\pm }+\mathrm{UD}C_{j\;=\;9}^{\pm }\;\times \;x_{j\;=\;9}^{\pm }\underset{\approx }{\leq }\mathrm{MG}I^{\pm } \end{array}$$
where MGI = maximum government investment in Liannan county; “$\underset{\approx }{\leq }$” means fuzzy less than.

(ii) Grain input–output constraints

Grain security is the main issue of Liannan County. In the model, the grain is produced by cultivated land. The grain production should afford the demand of Liannan County:(4)$$\sum_{k\;=\;1}^{4}\left(\mathrm{UG}P_{j\;=\;1,\;k}^{\pm }\;\times \;x_{j\;=\;1,\;k}^{\pm }\right)\underset{\approx }{\geq }\mathrm{DG}P^{\pm }$$
where UGP = unit grain production from cultivated land; DGP = demand grain production in Liannan county; “$\underset{\approx }{\geq }$” means fuzzy greater than.

(iii) Water production input–output constraints

In Liannan county, water production should afford the demand in this area and satisfy the demand of industries. In the model, water production is provided by water land:(5)$$\mathrm{UW}P_{j\;=\;7}^{\pm }\;\times \;x_{j\;=\;7}^{\pm }\underset{\approx }{\geq }\mathrm{DW}P^{\pm }$$
where UWP = unit water production from water land; DWP = demand water production in Liannan county.

(iv) Available water consumption constraints

All land use need water; water supply comes from the rivers and lakes in Liannan county. In the model, the water consumption of all types of land use should not exceed the available water supply:(6)$$\sum_{k\;=\;1}^{4}\sum_{j=1}^{6}\left(\mathrm{UW}C_{j\;,\;k}^{\pm }\;\times \;x_{j,\;k}^{\pm }\right)+\sum_{j=8}^{9}\left(\mathrm{UW}C_{j}^{\pm }\;\times \;x_{j}^{\pm }\right)\underset{\approx }{\leq }\mathrm{AW}S^{\pm }$$
where UWC = unit water consumption of land use (*j* = 1–6, 8, 9); AWS = available water supply in Liannan county.

(v) Available electricity power consumption constraints

Similarly, all types of land use need electricity power; the total electrical power consumption should not exceed the available supply capacity:(7)$$\sum_{k\;=\;1}^{4}\sum_{j=1}^{6}\left(\mathrm{UE}C_{j,\;k}^{\pm }\;\times \;x_{j,\;k}^{\pm }\right)+\sum_{j=8}^{9}\left(\mathrm{UE}C_{j}^{\pm }\;\times \;x_{j}^{\pm }\right)\underset{\approx }{\leq }\mathrm{AE}S^{\pm }$$
where UEC = unit electric power consumption of all types of land use; AES = available electric power supply in Liannan county.

### 4.3. Social Constraints

(i) Land Carrying Capacity constraints

In Liannan county, the Land Carrying Capacity (LCC) is limited; the maximum population in a unit area should not exceed maximum LCC in a unit area:(8)$$PP/\left(\sum_{k\;=\;1}^{4}\sum_{j=1}^{6}x_{j,\;k}^{\pm }+\sum_{j=7}^{9}\;x_{j}^{\pm }\right)\underset{\approx }{\leq }\mathrm{MLC}C^{\pm }$$
where PP = planning population; MLCC = maximum LCC in a unit area in Liannan county.

(ii) Available labor constraints

In Liannan county, all industries related to land use need labor; planning labor in Liannan county should not exceed the available labor:(9)$$\sum_{k\;=\;1}^{4}\sum_{j=1}^{6}\left(PLU_{j\;,\;k}^{\pm }\;\times \;x_{j,\;k}^{\pm }\right)+\sum_{j=7}^{9}\left(PLU_{j}^{\pm }\;\times \;x_{j}^{\pm }\right)\underset{\approx }{\leq }AL^{\pm }$$
where PLU = planning labor in a unit land area; AL = available labor in Liannan county.

### 4.4. Land Suitability Constraint

Land suitability assessment is indispensable for land-use allocation. Land areas of some types of land use (*j* = 1–6) should be in accordance with the results of land suitability assessment (water land, landfill, and unused land do not need land suitability assessment):(10)$$\sum_{j=1}^{6}x_{j}^{\pm }\underset{\approx }{\leq }\sum_{j=1}^{6}\mathrm{HS}L_{j}^{\pm }$$
where HSL = highly suitable land areas for land-use *j*.

### 4.5. Environmental Constraints

(i) Wastewater treatment capacity constraints

In the model, wastewater produced by some land types (*j* = 1–6) should not exceed the wastewater treatment capacity in Liannan county:(11)$$\sum_{k\;=\;1}^{4}\sum_{j=1}^{6}\left(\mathrm{WD}F_{j\;,\;k}^{\pm }\;\times \;x_{j,\;k}^{\pm }\right)\leq \mathrm{WP}C^{p}$$
where WDF = wastewater discharging factors of some types of land use (*j* = 1–6); WPC = wastewater treatment plant capacity in Liannan county; *p* = probability of violating the constraints of environmental capacities, and *p* ∈ [0,1].

(ii) Solid waste treatment capacity constraints

Similarly, solid waste produced by some land types (*j* = 1–6) should not exceed the solid waste treatment capacity and solid waste handle abilities of the landfill in Liannan county:(12)$$\sum_{k\;=\;1}^{4}\sum_{j=1}^{6}\left(\mathrm{SD}F_{j\;,\;k}^{\pm }\;\times \;x_{j,\;k}^{\pm }\right)-\mathrm{SH}L_{j\;=8}^{\pm }\;\times \;x_{j=8}^{\pm }\leq \mathrm{ST}C^{p}$$
where SDF = solid waste discharging factors of some types of land use (*j* = 1–6); SHL = solid waste handled by unit area of landfill; STC = solid waste treatment plant capacity (except landfill) in Liannan county.

(iii) Air pollutant discharge capacity constraints

Similarly, air pollutants produced by some land types (*j* = 1–6) should not exceed the air pollutant discharge capacity in Liannan county:(13)$$\sum_{k\;=\;1}^{4}\sum_{j=1}^{6}\left(\mathrm{AD}F_{j\;,\;k}^{\pm }\;\times \;x_{\;j,\;k}^{\pm }\right)\leq \mathrm{AD}C^{p}$$
where ADF = air pollutant discharge factors of some types of land use (*j* = 1–6); ADC = air pollutant discharge capacity in Liannan county (the air pollutant in this model is dust).

### 4.6. Ecological Constraints

(i) Available Soil Erosion constraints

Soil Erosion (SE) must be considered in cultivate land-use planning. In the model, we should consider the speed and impacts of cultivated land SE; the planning cultivate land SE area should not exceed available SE area in Liannan county:(14)$$\sum_{k\;=\;1}^{4}\left(\mathrm{SE}R_{j=1\;,\;k}^{\pm }\;\times \;x_{j=1,\;k}^{\pm }\right)\leq \mathrm{AS}E^{p}$$
where SER = SE rate of cultivated land; ASE = available cultivate land SE area in Liannan county.

(ii) Fertilizer consumption constraints

The cultivated land involves a key problem which is fertilizer application. Fertilizer supply is limited in Liannan county. In the model, fertilizer consumption should not exceed the maximum fertilizer consumption in Liannan county:(15)$$\sum_{k\;=\;1}^{4}\left(\mathrm{FC}U_{j=1\;,\;k}^{\pm }\;\times \;x_{j=1,\;k}^{\pm }\right)\leq \mathrm{MF}C^{p}$$
where FCU = fertilizer consumption in unit cultivate land; MFC = maximum fertilizer consumption in Liannan county.

### 4.7. Technical Constraints

(i) Total land areas constraints

(16)$$\sum_{k\;=\;1}^{4}\sum_{j=1}^{6}x_{j,\;k}^{\pm }+\sum_{j=7}^{9}x_{j}^{\pm }=\mathrm{TL}A^{\pm }$$

TLA = total land area of Liannan county.

(ii) Non-negative constraints

(17)$$x^{\pm }\geq 0$$

### 4.8. Parameters

Parameters of the proposed model are of four types: benefit/cost parameters, land-use suitability assessment parameters, social/economical parameters, and ecological/environmental parameters. Benefit parameters can be obtained from land price assessment, where the basic data could be acquired from the Overall Plan of Land Utilization of Liannan county (2010–2025) (Table 3); land-use suitability evaluation parameters can be obtained from Section 3; social/economical parameters can be obtained by data from the Statistical Yearbook of Liannan county (1995–2015) (Table 4); ecological/environmental parameters can be obtained fromEnvironmental Conditions Bulletin of Liannan county (1995–2015) (Table 5).

### 4.9. Model Solving

According to the model solution algorithm listed in formulas 18–24, using an interactive algorithm, the proposed model can be transformed into two deterministic sub-models, which correspond to the upper and lower bounds for the desired objective function value under different *p* levels. By arithmetic programming in software MATLAB 2016a, we can solve the model and obtain a series of land-use patterns, environmental emission scenarios, and ecological results. The framework of the proposed model is shown in Figure 6.

A general interval stochastic fuzzy programming (ISFP) model is coupled with interval mathematics programming, stochastic programming and fuzzy linear programming [54]:(18a)$$M\mathrm{ax}\;f^{\pm }\underset{\approx }{=}C^{\pm }x^{\pm }$$

Subject to:(18b)$$C^{\pm }x^{\pm }\underset{\approx }{\geq }b_{opt}^{\pm }$$
(18c)$$A_{i}^{\pm }x^{\pm }\underset{\approx }{\leq }b_{i}^{\pm }\;i=1,\;2,\;\ldots ,\;m,\;i\neq s$$
(18d)$$A_{s}^{\pm }x^{\pm }\underset{\approx }{\leq }b_{s}^{\left(p_{s}\right)}\;s=1,\;2,\;\ldots ,\;n,\;s\neq i$$
where *x* is a *n* × 1 alternative set; C is a 1 × *n* coefficient of an objective function; A*_i_* is a *m* × *n* matrix of coefficients of constraints; and *b_i_* is a *m* × 1 matrix (right-hand side (RHS)). “±” represents intervals; “$\underset{\approx }{=}$” represents fuzzy equality; “$\underset{\approx }{\geq }$” and “$\underset{\approx }{\leq }$” represents fuzzy inequality. *p_s_* denotes the probability that the constraints s are violated. $b_{s}^{\left(p_{s}\right)}$ represents corresponding values given the cumulative distribution function of *b_s_* and the probability of violating constraint *s* (*p_s_*).

On the basis of the principle of fuzzy flexible programming, let $\lambda^{\pm }$ value correspond to the membership grade of satisfaction for a fuzzy decision. Specifically, the flexibility in the constraints and fuzziness in the system objective, which are represented by fuzzy sets and denoted as “fuzzy constraints” and a “fuzzy goal”, are expressed as membership grades $\lambda^{\pm }$, corresponding to the degrees of overall satisfaction for the constraints/objective. Thus, model (1) can be converted to:(19a)$$M\mathrm{ax}\;\;\lambda^{\pm }$$

Subject to:(19b)$$C^{\pm }x^{\pm }\leq f_{opt}^{-}+(1-\lambda^{\pm })(f_{opt}^{+}-f_{opt}^{-})$$
(19c)$$A_{i}^{\pm }x^{\pm }\leq b_{i}^{-}+(1-\lambda^{\pm })(b_{i}^{+}-b_{i}^{-}),\;i=1,\;2,\;\ldots ,\;m,\;i\neq s$$
(19d)$$A_{s}^{\pm }x^{\pm }\underset{\approx }{\leq }b_{s}^{\left(p_{s}\right)}\;s=1,\;2,\;\ldots ,\;n,\;s\neq i$$
(19e)$$x^{\pm }\geq 0$$
(19f)$$0\leq \lambda^{\pm }\leq 1$$
where $f_{opt}^{+}$ and $f_{opt}^{-}$ denote the upper and lower bounds of the objective’s aspiration level as designated by decision makers; $\lambda^{\pm }$ denotes the control decision variable corresponding to the degree (membership grade) to which $x^{\pm }$ solution fulfills the fuzzy objective or constraints. Model (2) can be solved through a two-step method, where a sub-model corresponding to $\lambda^{-}$ is first formulated and solved. In the second step, the other sub-model corresponding to $\lambda^{+}$ can then be formulated supported by the solution of the first sub-model. If $b_{i}^{\pm }$ ≥ 0 and $f^{\pm }$ ≥ 0, the sub-model corresponding to $\lambda^{-}$ can be formulated as follows:(20a)$$M\mathrm{ax}\;\;\lambda^{-}$$

Subject to:(20b)$$\sum_{j=1}^{k_{1}}C_{j}^{+}x_{j}^{+}+\sum_{j=k_{1}+1}^{n}C_{j}^{+}x_{j}^{-}\leq f_{opt}^{-}+(1-\lambda^{-})(f_{opt}^{+}-f_{opt}^{-})$$
(20c)$$\sum_{j=1}^{k_{1}}{\left|a_{ij}\right|}^{-}\mathrm{Sign}(a_{ij}^{-})x_{j}^{+}+\sum_{j=k_{1}+1}^{n}{\left|a_{ij}\right|}^{+}\mathrm{Sign}(a_{ij}^{+})x_{j}^{-}\leq b_{i}^{-}+(1-\lambda^{-})(b_{i}^{+}-b_{i}^{-}),\;\forall i$$
(20d)$$\sum_{j=1}^{k_{1}}{\left|a_{sj}\right|}^{-}\mathrm{Sign}(a_{sj}^{-})x_{j}^{+}+\sum_{j=k_{1}+1}^{n}{\left|a_{sj}\right|}^{+}\mathrm{Sign}(a_{sj}^{+})x_{j}^{-}\leq b_{s}^{p_{s}},\;\forall s,\;s\neq i$$
(20e)$$x_{j}^{-}\geq 0,\;j=1,\;2,\;\ldots ,\;k_{1}$$
(20f)$$x_{j}^{+}\geq 0,\;j=k_{1}+1,\;k_{1}+2,\;\ldots ,\;n$$
(20g)$$0\leq \lambda^{-}\leq 1$$
where Sign is a signal function, which is defined as:$$S\mathrm{ign}(x^{\pm })=\left\{ \begin{array}{l} 1,\;if\;x^{\pm }\geq 0\\ -1,\;if\;x^{\pm }\leq 0 \end{array}\right.$$

Let $x_{j\;opt}^{+}$ (*j* = 1, 2, …, *k*_1_) and $x_{j\;opt}^{-}$ (*j* = *k*_1_ + 1, *k*_1_ + 2, …, *n*) be solutions of sub-model (3). Then, the second sub-model corresponding to $\lambda^{+}$ can be formulated supported by the solution of sub-model (4):(21a)$$M\mathrm{ax}\;\;\lambda^{+}$$

Subject to:(21b)$$\sum_{j=1}^{k_{1}}C_{j}^{-}x_{j}^{-}+\sum_{j=k_{1}+1}^{n}C_{j}^{-}x_{j}^{+}\leq f_{opt}^{-}+(1-\lambda^{+})(f_{opt}^{+}-f_{opt}^{-})$$
(21c)$$\sum_{j=1}^{k_{1}}{\left|a_{ij}\right|}^{+}\mathrm{Sign}(a_{ij}^{+})x_{j}^{-}+\sum_{j=k_{1}+1}^{n}{\left|a_{ij}\right|}^{-}\mathrm{Sign}(a_{ij}^{-})x_{j}^{+}\leq b_{i}^{-}+(1-\lambda^{+})(b_{i}^{+}-b_{i}^{-}),\;\forall i$$
(21d)$$\sum_{j=1}^{k_{1}}{\left|a_{sj}\right|}^{+}\mathrm{Sign}(a_{sj}^{+})x_{j}^{-}+\sum_{j=k_{1}+1}^{n}{\left|a_{sj}\right|}^{-}\mathrm{Sign}(a_{sj}^{-})x_{j}^{+}\leq b_{s}^{p_{s}},\;\forall s,\;s\neq i$$
(21e)$$x_{j\;opt}^{+}\geq x_{j}^{-}\geq 0,\;j=1,\;2,\;\ldots ,\;k_{1}$$
(21f)$$x_{j}^{+}\geq x_{j\;opt}^{-},\;j=k_{1}+1,\;k_{1}+2,\;\ldots ,\;n$$
(21g)$$0\leq \lambda^{+}\leq 1$$

Let $x_{j\;opt}^{-}$ (*j* = 1, 2, …, *k*_1_) and $x_{j\;opt}^{+}$ (*j* = *k*_1_ + 1, *k*_1_ + 2, …, *n*) be the solutions of sub-model (4). Thus, we can obtain the interval solutions as follows:(22a)$$\lambda_{opt}^{\pm }=\left[\lambda_{opt}^{-},\;\lambda_{opt}^{+}\right]$$

(22b)$$x_{j\;opt}^{\pm }=\left[x_{j\;opt}^{-},\;x_{j\;opt}^{+}\right],\;\forall j$$

Then, the optimized objective $f_{opt^{\prime }}^{-}$ and $f_{opt^{\prime }}^{+}$ can be calculated as follows:(23a)$$f_{opt^{\prime }}^{-}=\sum_{j=1}^{k_{1}}C_{j}^{-}x_{j}^{-}+\sum_{j=k_{1}+1}^{n}C_{j}^{-}x_{j}^{+}$$
(23b)$$f_{opt^{\prime }}^{+}=\sum_{j=1}^{k_{1}}C_{j}^{+}x_{j}^{+}+\sum_{j=k_{1}+1}^{n}C_{j}^{+}x_{j}^{-}$$

Thus, we have:(24)$$f_{j\;opt^{\prime }}^{\pm }=\left[f_{opt^{\prime }}^{-},\;f_{opt^{\prime }}^{+}\right],\;\forall j$$

## 5. Result Analysis

### 5.1. Optimized Land-Use Patterns

Here, we report the interval results from the proposed model under different *k* levels (*p* = 0.01) as follows. Optimized land-use patterns have been obtained under different scenarios. Under a high land suitability level (*k* = 1), optimized areas of cultivate land, garden land, forest land, grassland, construction land, transportation land, r water land, landfill and unused land are (10,825,11,958) hectare, (378,390) hectare, (102,469,113,695) hectare, (4203,4504) hectare, (3268,3891) hectare, (550,580) hectare, (1789,1890) hectare, (112,125] hectare and (550,600) hectare; similarly, under *k* = 2, the values will change to (10,750,11,210) hectare, (360,384) hectare, (102,411,113,214) hectare, (4106,4458) hectare, (3120,3785) hectare, (512,564) hectare, (1789,1890) hectare, (112,125) hectare and (550,600) hectare; when *k* = 3, land areas will be (10,213,10,650) hectare, (342,365) hectare, (97,290,107,553) hectare, (3901,4235) hectare, (2964,3596) hectare, (486,536) hectare, (1789,1890) hectare, (112,125) hectare and (550,600) hectare; when *k* = 4, the results are (9702,10,117) hectare, (325,347) hectare, (92,426,102,176) hectare, (3706,4023) hectare, (2816,3416) hectare, (462,509) hectare, (1789,1890) hectare, (112,125) hectare and (550,600) hectare. From the results, we can see that the model generates a series of interval results of land-use patterns. The interval results provide two extreme values for each result, which means the lower bound value and the upper bound value of the variables. Interval values do not give the distribution of the variables and objective but can reflect the amplitude of variation. Thus, the interval results could describe the various characters and can effectively support scenario analysis because various planning schemes can be generated by selecting a random value between the lower bound and the upper bound according to the demand. To generate detail and a common land-use planning scheme for change trend analysis, we can select the average value of the intervals.

### 5.2. Relationship between Land Suitability Level and System Benefit

Figure 7 shows the relationship between the land suitability level and NBL. When *k* = 1, NBL is (8.98, 15.14) × 10^9^ CNY; when *k* = 2, NBL will change to (6.23, 7.16) × 10^9^ CNY; when *k* = 3, NBL will reduce to (5.97,6.14) × 10^9^ CNY; and if *k* = 4, NBL is only (4.54, 5.94) × 10^9^ CNY. The results indicate that the land suitability level has a noteworthy influence on NBL. The land suitability level in Liannan county is influenced by many factors (see Section 4). The features of these factors will have an impact on all types of land use; therefore, the benefit from the land-use system will be influenced by these factors which respect the land suitability levels. The quantitative results from the optimization model could help land managers to give an exact insight into the relationship between land suitability level and benefit from land-use system.

### 5.3. Tradeoff between Economic Development and Environment Capacity

Figure 8 shows that NBL and the *p* levels have a relationship of positive correlation. When *p* = 0.01, NBL is (6.12,7.02) × 10^9^ CNY; when *p* = 0.05, NBL will change to (9.02,11.05) × 10^9^ CNY; when *p* = 0.10, NBL will rise to (12.68,19.24) × 10^9^ CNY; and when *p* = 0.15, NBL will be (22.91,30.57) × 10^9^ CNY. *p* level represents the probability of violating the ecological environment constraints. Any change in *p* level would yield different waste management capacities and thus results in different land-use patterns and different system benefits. From the results, if the environmental risk increases at 10%, the system benefit will increase at 3.0 × 10^9^ CNY in Liannan county.

### 5.4. Fuzzy Relationship between Objective and Constraints

The results also indicate that the optimized *λ* values are in the range of (0.36, 0.88). *λ* represents the possibility of satisfying all objectives and constraints under the given system conditions. The solutions correspond to conservative strategies when their *λ* values tend toward the lower bound; in comparison, the solutions become more optimistic when their *λ* values tend toward the upper bound. The relationship between *λ* values and NBL is shown in Figure 9. From this figure, we can see that *λ* values and NBL have a relationship of positive correlation. The optimization model is to achieve a maximized satisfaction degree (*λ* value) for system objectives and constraints under uncertainty. Under *λ* = 0.36, NBL will be (6.04,6.57) × 10^9^ CNY; while under *λ* = 0.88, NBL will be (8.98,15.14) × 10^9^ CNY. The *λ* values indicate the tradeoff between system benefit and all the constraints. Lower *λ* values would guarantee all the requirements are met, resulting in a stricter constraint and a lower system benefit; in comparison, higher *λ* values lead to more flexible constraints and a higher system benefit. For example, higher available electrical power, water, and soil erosion correspond to higher *λ* values and give a higher system benefit.

## 6. Discussion

The study of the land-use allocation model will be endless because of the complexity of the land-use system. Some limitations of the current study may be the direction for us to continue our research in the future. First, our study case comes from a county in China, which may have unique characteristics in some areas, such as mountainous terrain, large ethnic groups in the population, and so on. The proposed model has been successfully applied to a land-use allocation problem at a county scale; however, the application of this model to larger scales including more diverse regional differences such as watershed, regional and national scale needs to be studied. Second, due to the limitations of data sources, we did not consider climate change in the design of the model, and the factors related to environmental services were not considered adequately. Based on the proposed model, we will plan to further explore new design options, which will take into account climate change, environmental services, and ecological factors. Third, a multi-objective programming method could be coupled into the proposed model so as to solve multi-objective optimization problems; and the proposed model could be embedded to an intelligent model (such as the cellular automata model, multi-agent model, artificial immune system, ant colony model, and so on) so as to handle more complex land-use allocation problems. Last but not least, another possible area of future work would be to develop the proposed model into the corresponding software and improve accessibility through a user-friendly interface.

## 7. Conclusions

Previous land-use allocation models have two main problems: one is the inability to reflect the uncertainties in the land-use system [54,55], and the other is the lack of comprehensiveness and diversity of the constraints considered in the model [56,57]. In this study, we proposed an interval stochastic fuzzy programming mixed land suitability evaluation method-based land-use allocation model for land-use planning and the ecological environment management of Liannan county. This model is based on the interval stochastic fuzzy programming model and a land suitability evaluation method. Compared to the model in the previous studies, the model proposed in this study can effectively handle uncertainties expressed as discrete intervals, probabilities and fuzzy sets and thus can effectively support a policy analysis, tradeoff analysis and uncertain quantitative analysis. Furthermore, the model considers various economic, social, environmental and ecological factors in the land-use system which can give a series of land-use patterns, system benefits and ecological environmental protection strategies for sustainable development at a typical county level.The proposed optimization has been applied to a land-use planning practice in Liannan county and has obtained a series of results. These results can effectively support the government and land decision makers to formulate appropriate policies for land-use planning and ecological environmental management in Liannan county. First, with the growth of the population of Liannan County, the cultivated land is decreasing, and the reserve land is not sufficient. The proposed model is conducive to the local authorities to develop policies to protect cultivated land and to rationally determine the amount of cultivated land in different regions. Second, with the current Chinese government’s increasing emphasis on the ecological environment, the proposed model can support Liannan county’s policy development for environmental protection and improvement. Third, with the need for the further development of Liannan county, the proposed model can also support the formulation of policies to optimize land use and achieve sustainable development. Furthermore, this case study has proved the effectiveness, superiority, and practicability of the proposed model. The model may be applied to other regional scales when inputting corresponding parameters and conditions.

## Figures and Tables

**Figure 1 ijerph-16-04124-f001:**
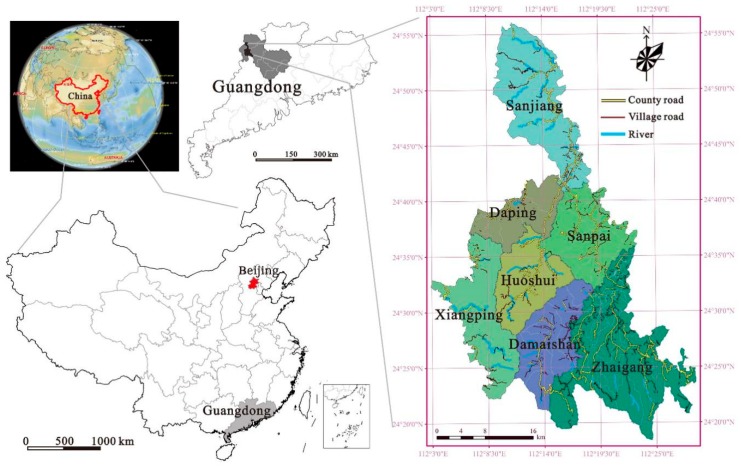
The study area.

**Figure 2 ijerph-16-04124-f002:**
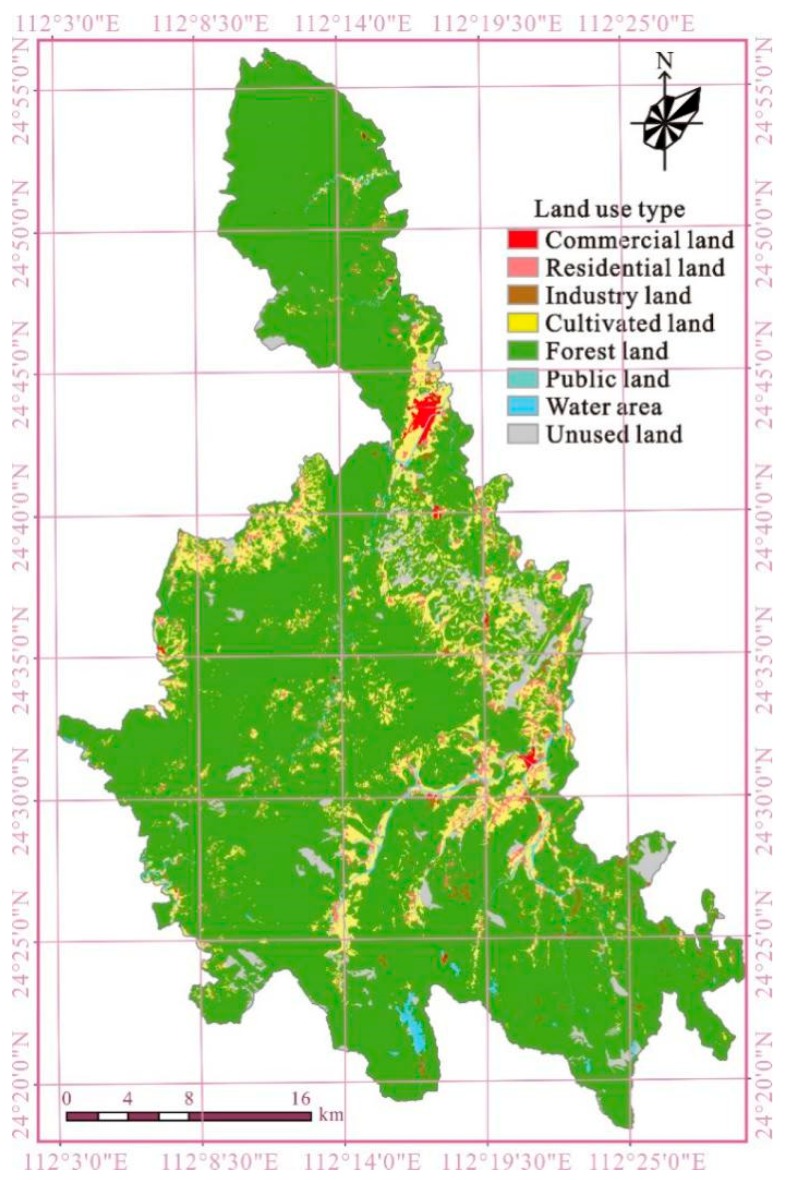
Current land-use classification.

**Figure 3 ijerph-16-04124-f003:**
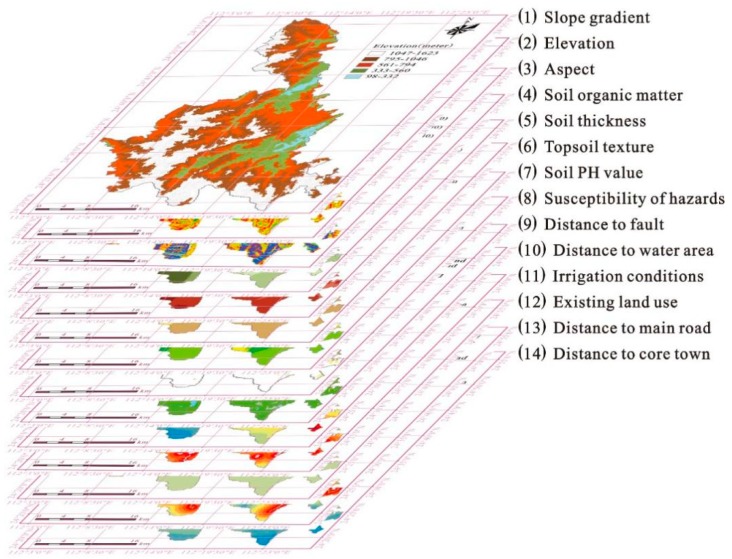
Overlap analysis of suitability factors.

**Figure 4 ijerph-16-04124-f004:**
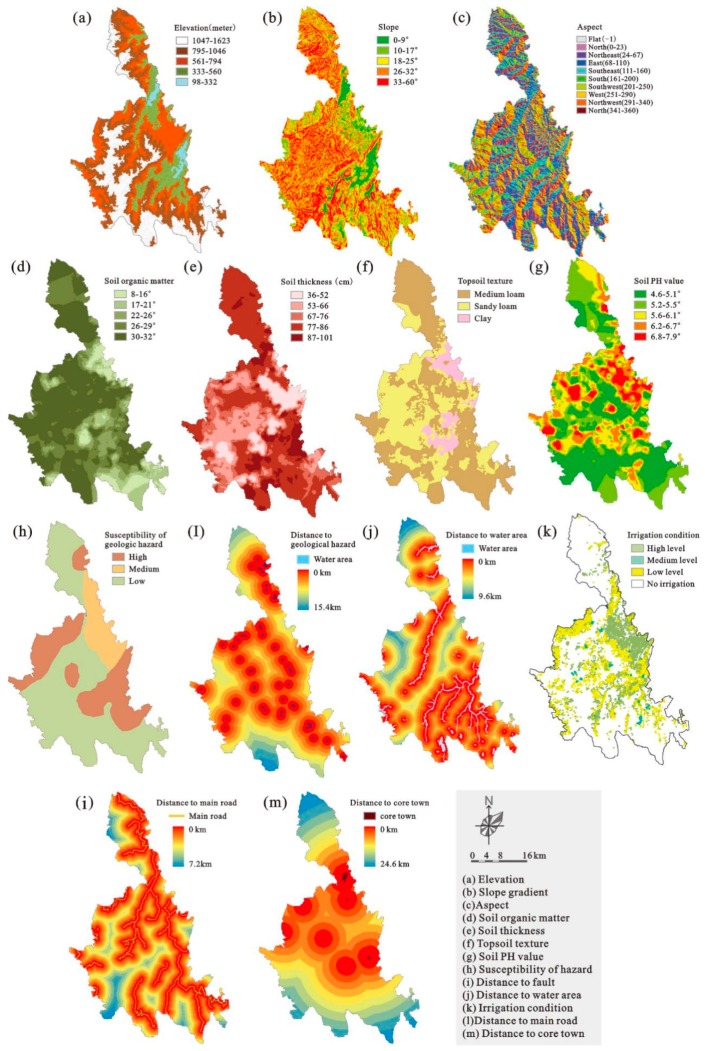
Details of Figure 3: Land suitability assessment for each factor (exciting land use see Figure 3).

**Figure 5 ijerph-16-04124-f005:**
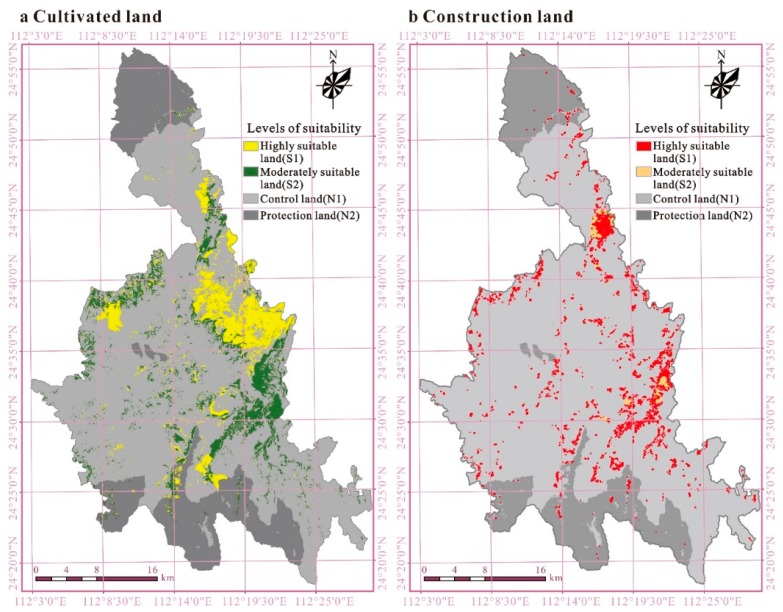
Suitability assessment maps for different land-use categories.

**Figure 6 ijerph-16-04124-f006:**
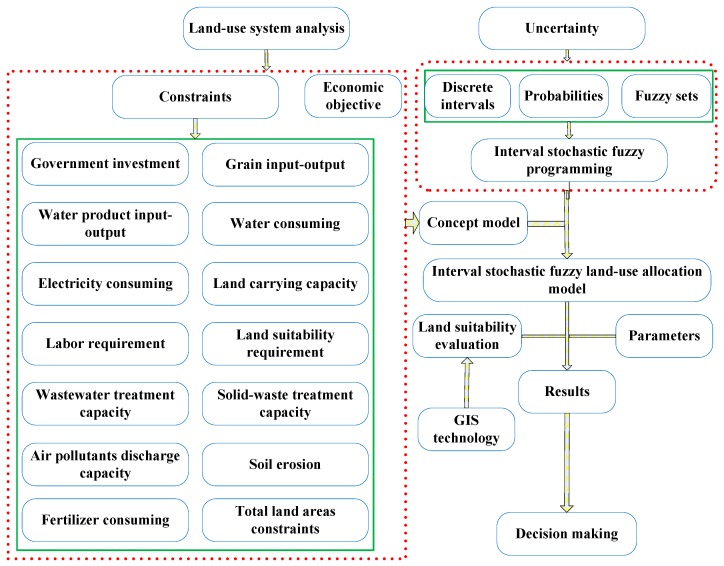
Framework of the proposed model.

**Figure 7 ijerph-16-04124-f007:**
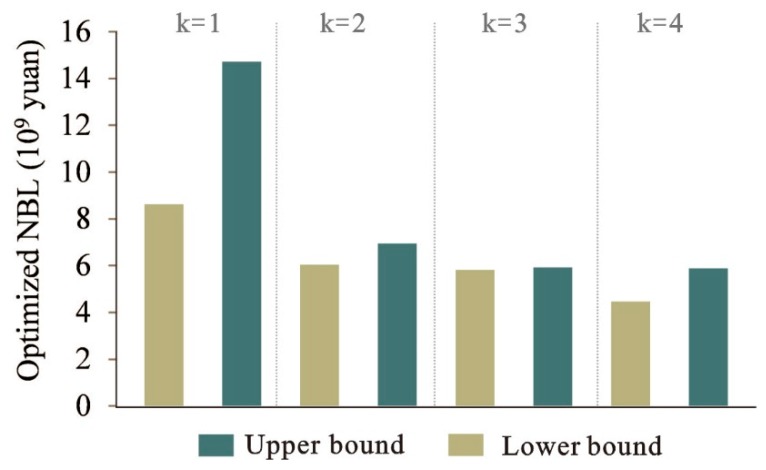
Relationship between land suitability level and system benefit.

**Figure 8 ijerph-16-04124-f008:**
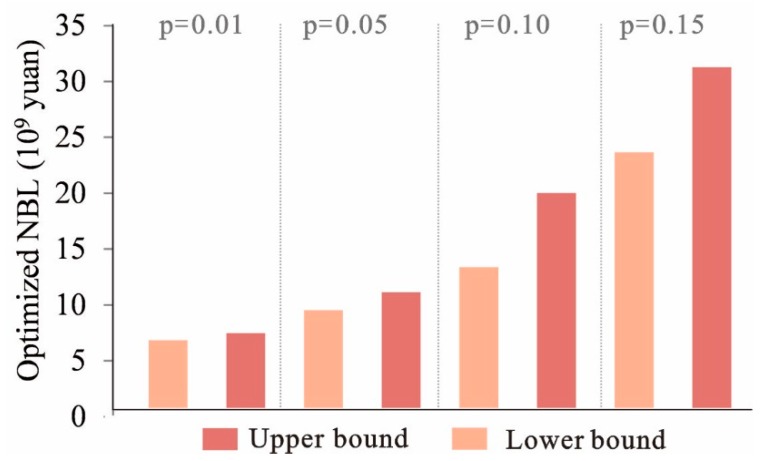
Tradeoff between economic development and environmental capacity.

**Figure 9 ijerph-16-04124-f009:**
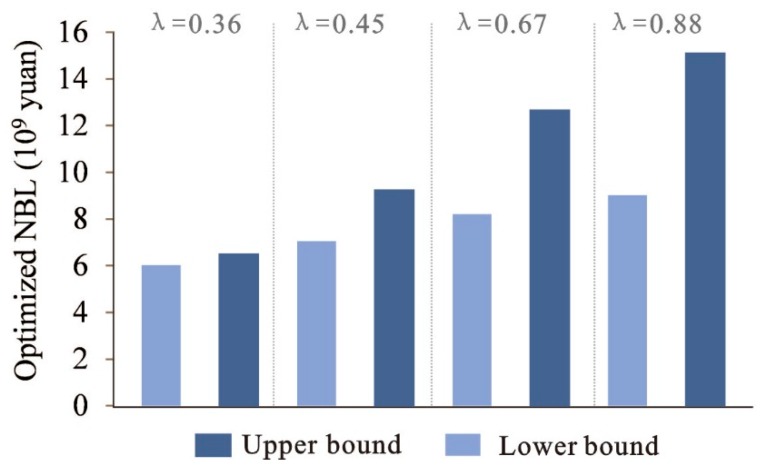
Fuzzy relationship between objective and constraints.

**Table 1 ijerph-16-04124-t001:** Selection of suitability factors for different land-use categories and their weights.

Physical Characteristics	Suitability Rating Score	S1 100–90	S2 90–80	S3 80–60	N1 60–30	N2 30
Cultivated land						
Topography	Slope gradient (°) (0.5)	<1.0 (100)	1.0–2.0 (90)	2.0–5.0 (80)	5.0–15.0 (40)	>15.0 (0)
	Elevation (meter) (0.3)	<30 (90)	30–50 (80)	50–100 (70)	100–200 (50)	>200 (10)
	Aspect (0.2)	Flat (100)	135–225 (90)	90–135,225–270 (80)	45–90, 270–315 (60)	0–45, 315–360 (50)
Ground conditions	Soil organic matter (%) (0.3)	>3.0 (100)	2.0–3.0 (90)	1.0–2.0 (80)	0.6–1.0 (60)	<0.6 (50)
	Soil thickness (centimeter) (0.3)	>100 (100)	60–100 (90)	-	30–60 (60)	<30 (30)
	Topsoil texture (0.2)	Medium loam (100)	Heavy/light loam (90)	Sandy loam (80)	Clay (60)	-
	Soil PH value (0.2)	6.0–7.9 (100)	5.5–6.0 (90)	5.0–5.5,7.9–8.5 (60)	4.5–5.0(50)	<4.5, >8.5 (30)
Geological hazards	Susceptibility of hazards (0.5)	Low (100)	Medium (90)	-	High (60)	-
	Distance to fault (meter) (0.5)	>300 (100)	100–300 (90)	50–100 (60)	30–50 (30)	<30 (10)
Hydrology conditions	Distance to water area (kilometer) (0.5)	0–0.5 (100)	0.5–1.0 (90)	1.0–2.0 (60)	2.0–5.0 (50)	>5.0 (30)
	Irrigation conditions (0.5)	High (100)	Medium (90)	Low (60)	No (30)	-
Commercial land, residential land, and industrial land						
Topography	Slope gradient (°) (0.5)	<1.0 (100)	1.0–2.0 (90)	2.0–5.0 (80)	5.0–15.0 (40)	>15.0 (0)
	Elevation (meter) (0.3)	<30 (90)	30–50 (80)	50–100 (70)	100–200 (50)	>200 (10)
	Aspect (0.2)	Flat	135–225	90–135, 225–270	45–90, 270–315	0–45, 315–360
Ground conditions	Existing land-use (0.6)	Construction land (100)	Cultivated land (90)	-	Forest land (40)	Water area (30)
	Topsoil texture (0.4)	Medium loam (100)	Heavy/light loam (90)	Sandy loam (80)	Clay (60)	-
Geological hazards	Susceptibility of hazards (0.5)	Low (100)	Medium (90)	-	High (60)	-
	Distance to fault (meter) (0.5)	>300 (100)	100–300 (90)	50–100 (60)	30–50 (30)	<30 (10)
Spatial location	Distance to main roads (meter) (0.5)	<100 (95)	100–300 (85)	300–800 (70)	800–1500 (40)	>1500 (10)
	Distance to core towns (0.5)	0–1.0 (100)	1.0–2.0 (90)	2.0–5.0 (60)	5.0–10.0(50)	>10.0 (30)

Note: Class S1—highly suitable: land having no significant limitations for sustained applications. Class S2—moderately suitable: land having limitations that in the aggregate are moderately severe for sustained application. Class S3—marginally suitable: land with limitations that in the aggregate are severe for sustained application. Class N1—currently not suitable: land that has qualities that appear to preclude sustained use. Class N2—permanently not suitable [49].

**Table 2 ijerph-16-04124-t002:** Descriptions of the symbols in the objective function.

Symbol	Meaning	Symbol	Meaning
NBL	Objective function, which means the net benefit from land-use system	UWTC	Unit wastewater-tackling cost of land-use types *j* = 1–7
±	Discrete interval values	USTC	Unit solid waste-tackling cost of land-use types *j* = 1–7
$\underset{\approx }{=}$	Fuzzy equal	UMC	Unit maintenance cost of landfill
*x*	Independent variables, which means the land areas of each land use	UDC	Unit developing costs of unused land.
*b*	Unit benefit of land use types *j* = 1–7	UESC	Unit electric power-supply cost of land-use types *j* = 1–7
*j*	Type of land use, where *j* = 1 for cultivate land, *j* = 2 for garden land, *j* = 3 for forest land, *j* = 4 for grassland, *j* = 5 for construction land, *j* = 6 for transportation land, *j* = 7 for water land, *j* = 8 for landfill, and *j* = 9 for unused land	UGTC	Unit waste-gas-tackling cost of land use types *j* = 1–7; UWSC = unit water-supply cost of land use types *j* = 1–7
		*k*	Land suitability condition, where *k* = 1 for highly suitable (S1); *k* = 2 for moderately suitable (S2); *k* = 3 for control suitable (N1); *k* = 4 for non-suitable (N2)

**Table 3 ijerph-16-04124-t003:** Benefit parameters (CNY ¥/hectare).

Benefit Parameters	Unit	*k* = 1		*k* = 2		*k* = 3		*k* = 4	
		Lower Bound	Upper Bound	Lower Bound	Upper Bound	Lower Bound	Upper Bound	Lower Bound	Upper Bound
*b_j_* _=1_	10^6^	2.35	5.04	1.95	4.28	1.64	4.01	1.24	3.32
*b_j_* _=2_	10^3^	3.51	5.64	3.25	5.01	2.89	4.49	2.19	3.67
*b_j_* _=3_	10^3^	1.25	2.04	1.12	1.98	1.02	1.77	0.95	1.28
*b_j_* _=4_	10^3^	2.59	4.98	2.44	4.57	2.05	4.08	1.59	3.63
*b_j_* _=5_	10^6^	0.34	0.68	0.29	0.59	0.22	0.51	0.18	0.41
*b_j_* _=6_	10^6^	0.12	0.26	0.10	0.24	0.08	0.22	0.06	0.20
*b_j_* _=7_	10^3^	1.12	1.89	1.01	1.74	0.96	1.59	0.88	1.23

Note: CNY is the Chinese monetary unit.

**Table 4 ijerph-16-04124-t004:** Social/economic parameters.

Symbol	Lower Bound	Upper Bound	Symbol	Lower Bound	Upper Bound
MGI (10^9^ CNY)	2.84	3.25	Maximum Land Carrying Capacity (MLCC) (person/hectare)	2	4
UGP (ton/hectare)	200	300	PLU (person/hectare)	1	3
DGP (10^3^ ton)	25.64	49.27	AL (person)	89,568	102,563
UWP (ton/hectare)	15	25	WDF (ton/hectare)	15.67	39.85
DWP (ton)	1000	2000	SDF (ton/hectare)	294.54	495.27
UWC (10^3^ m^3^/hectare)	100	150	SHL (ton/hectare)	25.69	69.18
AWS (10^9^ m^3^)	2.05	3.95	ADF (kg/hectare)	0.29	0.65
UEC (10^3^ kwh/hectare)	2.57	5.29	SER (%)	8%	15%
AES (10^9^ kwh)	0.35	0.58	FCU (ton/hectare)	0.2	0.3
PP (person)	170,321	178,657	TUL (10^3^ hectare)	102.35	162.14

**Table 5 ijerph-16-04124-t005:** Ecological/environmental parameters under different *p* levels.

Ecological Environmental Capacity	*p* Level			
	*p* = 0.01	*p* = 0.05	*p* = 0.10	*p* = 0.15
WPC (10^6^ ton)	(1.89,4.02)	(2.64,5.69)	(3.89,7.24)	(5.06,10.67)
STC (10^3^ ton)	(33.65,58.19)	(45.31,70.24)	(59.34,88.95)	(77.84,98.16)
ADC (ton)	(38.25,77.26)	(49.67,88.54)	(60.34,99.27)	(80.64,122.98)
ASE (hectare)	(1000,1100)	(1200,1400)	(1400,1700)	(1600,2000)
MFC (10^3^ ton)	(2.02,3.65)	(2.68,4.06)	(4.68,7.89)	(8.99,12.33)

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
