# Peer review of "Land Suitability Evaluation and an Interval Stochastic Fuzzy Programming-Based Optimization Model for Land-Use Planning and Environmental Policy Analysis"

_ijerph, 2019, doi:10.3390/ijerph16214124_

Round 1

Reviewer 1 Report

This manuscript is well-organized and can be accepted for publication. However, Appendix A and Appendix B should be moved to the context of the manuscript. It is not necessarily to put them as the Appendices.

Author Response

Reviewer #1: Comment 1 “This manuscript is well-organized and can be accepted for publication. However, Appendix A and Appendix B should be moved to the context of the manuscript. It is not necessarily to put them as the Appendices.” Response: * Many thanks to the reviewer’s suggestion. We have deleted the Appendix A and Appendix B and moved their contents to the body of the revised manuscript. ***End Response***

Reviewer 2 Report

Compared with the similar studies (at least 6-7 papers) his team did before, I think this paper lack of creativity, also there are various drawbacks. 1、As you know the most important is programming model, so I am wondering why there are only these costs rather than other costs, such as neglecting the conversion cost among landscapes, also in your objective formula, X8 and X9 has no benefit, therefore in the results, their quantities are the minimum limits, however your results are not like this. 2、Some constraints are set quite unreasonably, e.g. The study area is open, and has energy exchange with external world, so how to determine the quantities of water, food, and electric power that can be provided to it? 3、I cannot understand why you set this county into several towns, in your results the optimization structure of different towns are not given, it is not hierarchical optimization. Additionally, you set the constraint HSL, why not the constraints of S2, N2 and so on. 4、How to get the interval size of uncertain variable, which is very important? 5、There are various uncertainties, why in your studies there are interval, fuzzy and probabilities, could your explain their difference with regard to uncertainties. 6、There are total area constraint, while in optimization result it is not found in Table 5. 7、Actually, you know in China the land use structure in planning is determined by upper government, while in your constraints there are lack of such planning regulation. Welcome to hear your feedback

Author Response

Reviewer #2:

Comment 1

“As you know the most important is programming model, so I am wondering why there are only these costs rather than other costs, such as neglecting the conversion cost among landscapes, also in your objective formula, X8 and X9 has no benefit, therefore in the results, their quantities are the minimum limits, however your results are not like this.”

Response:

*Many thanks to the reviewer for the in-depth comments. In the programming model, we have made a lot of assumptions (Some assumptions maybe not proper). First, any benefit or cost from landscapes is net benefit or net cost. Benefit from some landscapes such as landfill and unused land (X8 and X9) are very small. So benefit from these landscapes is neglected (it is an assumption). Second, As we know, the results of X8 and X9 are from the programming model. The landfill is limited by the ability to handle the solid-waste; therefore, when the ability to handle the solid-waste is not enough, the model will allocate some quantities to this type of landscape (landfill is used to handle solid-waste). That is why the results of landfill are not the minimum limits. Similarly, how many quantities the model allocates to unused land is determined by the development ability of the local government (which is limited by the government investment). Many thanks to the reviewer, we have been aware of some assumptions may not proper, we will study them and improve the model in the following research.

Comment 2

“Some constraints are set quite unreasonably, e.g. The study area is open, and has energy exchange with external world, so how to determine the quantities of water, food, and electric power that can be provided to it?”

Response:

*Many thanks to the reviewer’s insightful comments. Again, we set this constraint by some assumptions (maybe not proper). The food, water, and electric power consuming and local supply capacity of a city (In China) are stated by local statistical bureau. The true net external providing quantities can be calculated by these data. According to this, we used some forecasting models to forecast the providing quantities in the planning years. In fact, we don’t really know these constraints are right or wrong, it is just exploring. Thanks again for the insightful comments.

Comment 3

“I cannot understand why you set this county into several towns, in your results the optimization structure of different towns are not given, it is not hierarchical optimization. Additionally, you set the constraint HSL, why not the constraints of S2, N2 and so on.”

Response:

*Many thanks to the reviewer’s useful comments. First, it is really necessary to divide the county into several towns in our initial imagining, because every town’s situation is not the same which will lead to different parameters. Unfortunately, we can’t get the parameter data from every town. So the reviewer is sagacious, in this study, it is no use to divide the county into several towns. In the revised manuscript, we delete the dividing. Second, we only use the S1 results is because in S1 the land suitability condition is the best, which can get the best land-use allocation patterns. However, it is necessary to discuss the S2, N1, N2 situations which can provide more choices for land managers; but due to limited length of the manuscript, we didn’t discuss these scenes in this study.

Comment 4

“How to get the interval size of uncertain variable, which is very important?”

Response:

*Many thanks to the reviewer’s professional comments. The interval values of variables are come from the model computing by input interval parameters. These interval parameters come from forecasting models. By means of observing years of data, we can find the rough interval values of the original parameter. Some values may not change obviously. For these data, we set 85%-115% of the values as the interval original parameters. This is another assumption of this model. Many thanks to the reviewer’s careful review; we are attempting to ensure the accuracy of every parameter data.

Comment 5

“There are various uncertainties, why in your studies there are interval, fuzzy and probabilities, could your explain their difference with regard to uncertainties.”

Response:

*Many thanks to the reviewer’s insightful comments. First, our programming model is based on Zhou [1] (2015)’ land-use allocation model, his/her model considered intervals, fuzzy sets, and probabilities. We have improved his/her model by adding some constraints and GIS-based land-suitability assessment. The main reason we choose these three uncertainties is the fact that these three uncertainties really exist in land-use allocation system. However, there may be other uncertainties in the system as the review point out; we will continue to study other uncertainties in the land-use allocation system, many thanks to the reviewer’s valuable comments. Second, the differences between these three uncertainties are: Interval uncertainties are expressed as interval values. This type of uncertainty may exist in every mathematical value. Every values could not be measured completely accurate. However, we can use an interval value to express it. Probability is another uncertainty that could describe the likelihood that the event will occur; while fuzzy set could describe the uncertainty of concept connotation. Every uncertain is suitable for special parameters. For example, interval uncertainty is suitable for all parameters; fuzzy uncertainty is suitable for inequation or equation condition; probability uncertainty is suitable for environmental/ecological capacity. Many thanks to the reviewer’s professional and in-depth comments.

[1] Zhou, M., An interval fuzzy chance-constrained programming model for sustainable urban land-use planning and land use policy analysis. Land Use Policy 2015, 42, 479-491.

Comment 6

“There are total area constraint, while in optimization result it is not found in Table 5.”

Response:

*Many thanks to the reviewer’s careful concern. We established the model base on the assumption that the total area of the study area will not change in the planning period.

Comment 7

“Actually, you know in China the land use structure in planning is determined by upper government, while in your constraints there are lack of such planning regulation.”

Response:

*Many thanks to the reviewer’s insightful comments. Actually, in China, the land use structure in planning is determined by the upper government. The upper government will limit every land-use type by policy. Our model is to get some results that determined by just nature rules and technical means. Some constraints are limited by government policies, such as land suitability constraints, government investment constraints. Furthermore, the government of China makes many accounts of environment and ecology. The environmental and ecological constraints reflect these policies. The reviewer’s professional comments are appreciative. These comments really can help us to improve the quality of our study.

***End Response***

Reviewer 3 Report

This paper needs a major revision before it could be accepted. Although the application of fuzzy land use allocation is innovative, the weak readability significantly undermines its overall quality. The following points need to be addressed:

First, I am totally confused by line 80-83, “… involves various economic, social, ecological and environmental constraints, most of which are firstly put into the optimization process …”, are the authors proposing a new optimization process for the economic, social, ecological and environmental constraints? What is the meaning of ‘firstly’ here? I must highlight the large number of literature that has already explored these constraints for land use allocation study, such as Bertaud and Renaud (1997).

Bertaud, A. and Renaud, B., 1997. Socialist cities without land markets. Journal of urban Economics, 41(1), pp.137-151.

Second, I strongly recommend authors get help from a native speaker to improve the readability of this paper. Numerous typos and grammar mistakes gave me a hard time to follow this manuscript. For example, in line 30, is Liannan a 'country' or a county? Also it is better to replace the text from line 227 to line 239 with a table. The six equations of 10a-f should be combined. 

Third, authors needs to pay more attention to the usage of reference. For example, in line 181-182, authors should give a citation to the Delphi approach since they do not propose it here.

To summarize, a major revision is necessary before this paper could be accepted.

Author Response

Reviewer #3:

Comment 1

“First, I am totally confused by line 80-83, “… involves various economic, social, ecological and environmental constraints, most of which are firstly put into the optimization process …”, are the authors proposing a new optimization process for the economic, social, ecological and environmental constraints? What is the meaning of ‘firstly’ here? I must highlight the large number of literature that has already explored these constraints for land use allocation study, such as Bertaud and Renaud (1997).”

Response:

* Thank you very much for the reviewer's careful review and helpful reminder. Indeed, the use of “firstly” is not appropriate, we have corrected the corresponding statement as follows. “This model not only considers three uncertainties in the natural system but also involves various economic, social, ecological and environmental constraints, most of which are specifically put into the optimization process.”

Comment 2

“Second, I strongly recommend authors get help from a native speaker to improve the readability of this paper. Numerous typos and grammar mistakes gave me a hard time to follow this manuscript. For example, in line 30, is Liannan a 'country' or a county? Also it is better to replace the text from line 227 to line 239 with a table. The six equations of 10a-f should be combined.”

Response:

* Many thanks to the reviewer’s suggestion. The current language in this manuscript was revised by a native English-speaking editor.

* We corrected “country” to “county” in the revised manuscript.

* According to the reviewer's guidance, we listed the symbols and descriptions mentioned in the objective function in a new table. For details, please see Table 2 in the revised manuscript.

* The equations of 10a-f have been combined in the revised manuscript.

Comment 3

“Third, authors needs to pay more attention to the usage of reference. For example, in line 181-182, authors should give a citation to the Delphi approach since they do not propose it here.”

Response:

* Many thanks to for reviewer’s suggestion. We added the following reference in the revised manuscript: ”McKenna, H. P., The Delphi technique: a worthwhile research approach for nursing? Journal of advanced nursing 1994, 19, (6), 1221-1225.”

***End Response***

Reviewer 4 Report

Review of Zhang et al., Land suitability evaluation and an interval stochastic 3 fuzzy programming based optimization model for land use planning and environmental policy analysis

For International Journal of Environmental Research and Public Health

This paper should be further reviewed before publication to improve English, reduce the detail about Guangdong Province, give the case study more generality by expanding it beyond an economic analysis, and provide a proper discussion.

General

While I can follow the mathematics and the modelling, I am not an appropriate critical reviewer for these elements of the paper. My primary interests and expertise are in relation to policy development and the value and importance of and use planning as a process.

The model appears to be a viable option for land use planning, and is potentially better than many of the options available today. Its use of fuzzy logic is an interesting development that enables coping with the many unknowns always present when attempting to predict the future. The challenge for the authors is to create user-friendly software that will make the model accessible to planners. That development is the area where assumptions and background criteria become of critical importance, because every land use planning problem is unique. Some more specific comments on how to make the model accessible would be useful.

I found the case study to be uninteresting, primarily because it did not incorporate the two most important elements of environmental management facing planners today: climate change and environmental services. The example focuses entirely on issues related to planning for humans in the current world.  By assuming “business-as-usual”, the example is immediately outdated. This example is almost entirely an economic model, with some small consideration of social issues and reference to a couple of environmental issues of direct relevance to human enterprise. This is not an example that will encourage modern planners to use the model.

The authors note that the model is versatile, and can be adapted to incorporate any measurable parameter. I suggest that putting stronger elements of environmental management into the model will strengthen its validity and make the case study more convincing.

For me, there was too much information giving background to the case study. None of that information was used in the subsequent discussion. While I realise that it is included in order to explain the layering principles underlying the analysis, I think that GIS methodology is now well enough understood that the background detail about Guangdong Province is not necessary.

Specific

The English needs more editing to make the paper clearer. While there were many minor grammatical issues that were unimportant in terms of understanding, there were some areas where the key points were very difficult to follow. The authors should work with a knowledgeable editor to improve the writing. I note, however, that in many places the English in the paper is remarkably good and congratulate the authors on their achievement.

It is standard practice to report data in either a table, or the text, but not both. There is much repetition of tabled results in the text in Section 5 in particular, and that should be taken out.

I felt that the conclusions were more of a summary, and added little to the paper. The comments there were not really either discussion or conclusions. The authors might give some specific examples of how the model results could contribute to policy development.

The figures need to be improved for presentation purposes. Frame the bars, adjust the spacing between bars to make the bars more prominent, consult with the editors about use of colours and texture.

Recommendations:

Reduce the background information given about the case study.

Demonstrate how climate change and environmental services might be incorporated into the model.

Provide a discussion that explains why the model is useful and interesting, and shows how it might be used to contribute to policy development.

Author Response

Reviewer #4:

Comment 1

“The model appears to be a viable option for land use planning, and is potentially better than many of the options available today. Its use of fuzzy logic is an interesting development that enables coping with the many unknowns always present when attempting to predict the future. The challenge for the authors is to create user-friendly software that will make the model accessible to planners. That development is the area where assumptions and background criteria become of critical importance, because every land use planning problem is unique. Some more specific comments on how to make the model accessible would be useful.”

Response:

* Many thanks to the reviewer’s helpful suggestion, which guides our future work. According to the reviewer's comments, we modified Section 6 to add a lot of content. For details, please see “6. Concluding discussion” in the revised manuscript.

Comment 2

“I found the case study to be uninteresting, primarily because it did not incorporate the two most important elements of environmental management facing planners today: climate change and environmental services. The example focuses entirely on issues related to planning for humans in the current world.  By assuming “business-as-usual”, the example is immediately outdated. This example is almost entirely an economic model, with some small consideration of social issues and reference to a couple of environmental issues of direct relevance to human enterprise. This is not an example that will encourage modern planners to use the model.”

Response:

* Due to the limitations of data sources, we did not consider climate change in the design of the model, and the factors related to environmental services were not considered adequately. Based on the proposed model, we will plan to further explore new design options, which will take into account climate change, environmental services, and ecological factors.

* We explained the limitations mentioned above in the revised manuscript. For details, please see “6. Concluding discussion” in the revised manuscript. Many thanks to the reviewer’s constructive suggestions.

Comment 3

“The authors note that the model is versatile, and can be adapted to incorporate any measurable parameter. I suggest that putting stronger elements of environmental management into the model will strengthen its validity and make the case study more convincing.”

Response:

* Many thanks to the reviewer’s suggestion. Based on the implications of this suggestion, we further explain the shortcomings of our research and the focus of our work in the future. For details, please see “6. Concluding discussion” in the revised manuscript.

Comment 4

4-1:“For me, there was too much information giving background to the case study. None of that information was used in the subsequent discussion. While I realise that it is included in order to explain the layering principles underlying the analysis, I think that GIS methodology is now well enough understood that the background detail about Guangdong Province is not necessary.”

4-2:“Reduce the background information given about the case study.”

Response:

* Thank you very much for the reviewer's suggestion. After a discussion between the authors, we decided to delete the introduction of the study area in the manuscript about Guangdong Province. The following is the details of the deletion.

Guangdong is one of the most economically developed provinces in China [48]. Compared with other economically developed regions such as the Pearl River Delta in Guangdong province, Liannan economic development level is still relatively low, the total economic volume is small; the independent innovation ability is weak; the overall economic efficiency is not high enough, and the industrial structure needs to be further optimized. For example, the total annual GDP of Liannan in 2015 was 3.586 billion CNY, up by 7.3%, higher than the national level (6.9%), but lower than the average level of Guangdong province (8.0%) and Qingyuan city (8.4%). The proportion of agriculture, industry, service sectors of Liannan in 2015 was 15.9: 33.0: 51.1. The added value created by agriculture sector was 570 million CNY, up by 6.0%, and the contribution rate to GDP growth was 13.6%; the added value of the industry sector was 1.184 billion CNY, up by 1.6 %, and the contribution rate of GDP growth was 9.2%; the added value of service sector was 1.832 billion CNY, up by 13.8%, and the contribution rate of GDP growth was 77.3%. Per capita GDP of Liannan in 2015 was 26,853 CNY (4,311 U.S. dollars), an increase of 7.4%.”

Comment 5

“The English needs more editing to make the paper clearer. While there were many minor grammatical issues that were unimportant in terms of understanding, there were some areas where the key points were very difficult to follow. The authors should work with a knowledgeable editor to improve the writing. I note, however, that in many places the English in the paper is remarkably good and congratulate the authors on their achievement.”

Response:

* Many thanks to the reviewer’s suggestion. The current language in this manuscript was revised by a native English-speaking editor.

Comment 6

“It is standard practice to report data in either a table, or the text, but not both. There is much repetition of tabled results in the text in Section 5 in particular, and that should be taken out.”

Response:

* Many thanks to the reviewer’s suggestion. The repetition problem does exist 

in Section 5.1. Based on the reviewer's suggestion, we deleted Figure 6 and modified some of the textual representations in Section 5.1.

Comment 7

“I felt that the conclusions were more of a summary, and added little to the paper. The comments there were not really either discussion or conclusions. The authors might give some specific examples of how the model results could contribute to policy development.”

Response:

* Many thanks to the reviewer’s suggestion. We changed the subtitle of Section 6 to "Concluding discussion". Specifically, we have compared the model proposed in this study with the models in the previous literature and added policy implications for our research. For details, please see “6. Concluding discussion” in revised manuscript.

Comment 8

“The figures need to be improved for presentation purposes. Frame the bars, adjust the spacing between bars to make the bars more prominent, consult with the editors about use of colours and texture.”

Response:

* Many thanks to the reviewer’s suggestion. Under the guidance of the reviewer, we redrawn Figures 7, 8 and 9 in the revised manuscript. The new column charts illustrate our calculation results more compactly and effectively.

Comment 9

“Demonstrate how climate change and environmental services might be incorporated into the model.”

Response:

* Please see Response to Comment 2.

Comment 10

“Provide a discussion that explains why the model is useful and interesting, and shows how it might be used to contribute to policy development.”

Response:

* Please see Response to Comment 7.

***End Response***

Round 2

Reviewer 2 Report

It provides acceptable feedback to my doubts, so can be accepted in present form. 

Reviewer 3 Report

I am satisfied with the revision that addresses all my concerns.